# Antioxidant Supplementation on Male Fertility—A Systematic Review

**DOI:** 10.3390/antiox12040836

**Published:** 2023-03-30

**Authors:** Fotios Dimitriadis, Hendrik Borgmann, Julian P. Struck, Johannes Salem, Timur H. Kuru

**Affiliations:** 1Urology Department, School of Medicine, Faculty of Health Sciences, Aristotle University of Thessaloniki, 54124 Thessaloniki, Greece; 2Department of Urology, Faculty of Health Sciences Brandenburg, Brandenburg Medical School Theodor Fontane, 14476 Potsdam, Germany

**Keywords:** antioxidant, male fertility, supplementation

## Abstract

Our aim was to review the current literature regarding the effect of antioxidant supplementation (AS) on male fertility parameters, as AS is commonly used to treat male infertility due to the availability and affordability of antioxidants in many parts of the world. Materials and methods: PubMed, Medline, and Cochrane electronic bibliographies were searched using the modified Preferred Reporting Items for Systemic Reviews and Meta-Analyses (PRISMA) guidelines to evaluate studies on the benefit of antioxidant therapy on infertile men. Results were analyzed regarding the following aspects: (a) ingredient and dose; (b) potential mechanism of action and rationale for use; and (c) effect on various reported outcomes. Results: Thus, 29 studies found a substantial positive effect of AS on outcomes of assisted reproductive therapy (ART), WHO semen parameters, and live-birth rate. Carnitines, Vitamin E and C, N-acetyl cysteine, coenzyme Q10, selenium, zinc, folic acid, and lycopene were beneficial ingredients. Nevertheless, some studies did not show a substantial change in one or more factors. Conclusion: AS seems to have a positive effect on male fertility. Environmental factors may play an increasing role in fertility. Further studies are needed to determine the optimal AS combination and the influence of environmental factors.

## 1. Introduction

Infertility is defined as the failure to become pregnant despite frequent, unprotected intercourse for at least one year [1].

More than 80 million couples worldwide complain of infertility. Male infertility is a controversial issue worldwide for several, multi-factorial reasons. Further, 8 to 12% of all couples suffer from infertility [2], with a growing tendency. In this context, male infertility occurs in 30–50% of cases [3]. Varicocele, smoking, radiation, urinary tract infection, nutritional deficiencies, oxidative stress, and environmental factors are some reasons for male factors [4,5,6] but seem not to be completely understood in all aspects. Oxidative stress (OS) may play an increasingly important role in environmental factors when the production of reactive oxygen species (ROS) exceeds the body’s natural antioxidant neutralizing capacity [7,8].

The increase in ROS concentration may be due to environmental factors, such as high temperature, electromagnetic waves, air pollution, insecticides, alcohol consumption, obesity, and poor nutrition [9], all of which have shown an increasing trend in recent years. The latest evidence from 2022 supported the Mediterranean diet as a shield against male infertility and cancer risk induced by environmental pollutants [10]. The authors reported about a decline in male fertility worldwide due to environmental pollution. A report was recently published about molecular alterations and severe abnormalities in the spermatozoa of young men living in polluted areas [11].

Some evidence supports the role of ROS in male infertility [12,13,14,15]. Spermatozoa have a unique plasma membrane structure, containing significant levels of polyunsaturated fatty acids (PUFAs) improving membrane flexibility required for oocyte penetration. This membrane is unfortunately very vulnerable to attack by ROS [14,16]. The exact reaction seems to be a lipid peroxidation cascade that compromises membrane cell integrity, decreasing sperm motility and subsequently reducing fertility. Additionally, ROS leads to significant DNA damage [17].

Natural antioxidants in humans include vitamin C and E, superoxide dismutase, thioredoxin, and glutathione. The mentioned antioxidants are capable of neutralizing free radical activity and protect spermatozoa from ROS [7]. This results in a lower concentration of antioxidants in the semen of infertile men. This observation may explain the high levels of ROS in infertile men compared to fertile men [18]. A lot of improvement has been seen in the field of male infertility in recent decades. Sperm function tests, such as sperm DNA fragmentation (SDF) and measurements of OS, are widely used to provide a better understanding of true male fertility potential [19]. Table 1 shows the method of action of available antioxidants.

Semen analysis can show the following pathologies: asthenospermia (reduced motility), oligozoospermia (low concentration), and teratozoospermia (abnormal morphology), and the combination of all (WHO 2021 classification). According to the current WHO 2021 manual, standard values of sperm parameters are as follows: total sperm count 39 million per ejaculate or greater, sperm concentration = or > than 16 mio/mL, sperm vol. 1.4 mL or higher, progressive sperm motility of 30% or more, and normal morphology equal to or >4% [31].

In recent centuries, andrologists have paid the most attention to the effect of oxidative stress on male infertility and the role of oral AS in improving semen quality. Most of these studies report a positive correlation between AS and male fertility. However, some studies also yielded negative results and led to conflicting recommendations. With regard to the latest EAU guidelines, no clear recommendation can be made for AS [32].

This review investigates the benefit of AS in the latest studies on semen parameters, such as sperm concentration, morphology DNA damage, motility, and fertility rate, and tries to give an update on this topic.

## 2. Materials and Methods

### 2.1. Research Strategy

Our literature research was designed according to the modified Preferred Reporting Items for Systematic Reviews and Meta-Analyses (PRISMA) guidelines. Publication search of the PubMed, Medline, and Cochrane electronic bibliographies was conducted to find studies reporting about the benefits of antioxidants for male infertility. The research was performed using the following keywords: ‘antioxidants’, ‘semen parameters’, ‘male infertility’, ‘pregnancy rate’, ‘live birth rate’, and ‘sperm function’. Medical Subject Heading (MeSH) phrases included: (‘Antioxidants’(MeSH)) AND ‘Infertility, Male’(MeSH), (‘Antioxidants’(MeSH)) AND ‘Pregnancy Rate’(MeSH), (‘Antioxidants’(MeSH)) AND ‘Semen Analysis’(MeSH).

### 2.2. Study Selection

The list of articles received was evaluated by title and abstract by one author (TK). Relevant full articles were analyzed in detail. Review articles were also screened to find other suitable articles. Exclusion criteria were as follows: gender (females), species (other animals), and language (non-English). Data were then extracted, cross-checked, and verified. See Figure 1 for a PRISMA flowchart of study selection.

### 2.3. Outcome Measures

The results of the studies evaluated were as follows: mechanism of action (if indicated), AS type and dosage (if available), effects on basic semen parameters and extended sperm function tests, and live-birth rate. 

### 2.4. Results

Our literature search showed 622 papers, of which 531 were excluded based on title and/or abstract. The remaining 91 papers were reviewed in detail, with 50 articles identified as meeting the inclusion and exclusion criteria (Table 2). The most frequently studied AS and doses used were: vitamin C (500–1000 mg), vitamin E (400 mg), zinc (25–400 mg), carnitines (L-carnitine (LC) or L-acetyl-carnitine (LAC)) (500–1000 mg), co-enzyme Q10 (CoQ10; 100–300 mg), N-acetyl cysteine (NAC; 600 mg), selenium (Se) (200 mg), folic acid (0.5 mg), and lycopene (6–8 mg). The antioxidants that were used are shown in Table 3.

## 3. Discussion

In the following discussion section, we will discuss and demonstrate the most commonly used ASs and provide details of the studies evaluated. Not all antioxidants can be discussed in this section, as we have weighed their use in daily practice worldwide.

### 3.1. Vitamin E + Vitamin C

Ascorbic acid (vitamin C) is a water-soluble antioxidant acting as an important cofactor in hydroxylation and amidation reactions [20]. In combination with vitamin E, it plays a key role in collagen synthesis, proteoglycans, and intercellular matrix [55]. Vitamin C was found in large amounts in seminal plasma [56,57]. Higher doses of vitamin C intake result in higher concentrations in seminal plasma and may prohibit DNA damage [58].

Vitamin E is a fat-soluble antioxidant capable of neutralizing free radicals, thus protecting cell membranes from free radicals. It also prevents the lipid peroxidation cascade, thus improving the functions of other ASs [24]. Furthermore, Vitamin E has been shown to inhibit the production of ROS in infertile men [49].

Gerco et al. reported on an intervention study of infertile men [44]. The intervention group in this study received one gram of vitamin E and one gram of vitamin C. The extent of DNA damage was significantly reduced in the intervention group (*p* < 0.001) after two months. Nevertheless, no positive correlation was found between vitamin E and C treatment and the main semen parameters (motility and concentration.) In a second study by these authors, intracytoplasmic sperm injection (ICSI) and in vitro fertilization (IVF) show that the significant rate of sperm DNA damage leads to lower infertility rates by using the aforementioned AS [44].

Therefore, the authors stated that two months of treatment with one gram of vitamins C and E improved the ICSI success rate in patients with impaired sperm DNA damage and reduced the rate of DNA damage in these individuals.

Moslemi et al. reported on 690 infertile men with idiopathic OAT who received daily AS with selenium (200 μg) in addition to vitamin E (400 IU). Supplements were given for at least 100 days. The authors reported a 52.6% (362 cases) overall improvement in sperm motility, morphology, or both, and a percentage of 10.8% (75 cases) spontaneous pregnancy compared with no treatment [40].

### 3.2. Carnitine

L-carnitine (LC) (3-aminobutyric acid) occurs in humans and is also a vitamin metabolized in the human body. The involvement of LC is essential in intermediary metabolism, playing an important role in the formation of acyl carnitine esters [59]. High concentrations of LC in the human body occur in the epididymis, more than 2000-times higher than in serum [60,61]. The high level of LC in the epididymis is a result of a continuous secretory process in this organ [59]. The findings show a positive correlation of sperm movement and increased LC (epididymis) and L-acetyl (semen) [60].

To date, a few studies investigated the effect of L-carnitine AS. Lenzi et al. reported a double-blind controlled clinical trial on the effect of LC on male infertility. Thus, 60 infertile men with OAT were treated in an intervention and control group. In the intervention group, patients received 2 g/day LC and 1 g/day L-acetyl carnitine (LAC) for 6 months during the study. A positive association between LC and LAC and sperm motility was observed in the study population. Interestingly, this association was more significant with lower sperm motility at baseline sperm quality assessment [41].

Balercia et al. reported the effects of LC and LAC on sperm motility and total oxygen radical scavenging capacity (TOSC) [42]. This randomized, double-blind controlled trial reported 60 men with idiopathic OAT. After six months, patients treated with LC and LAC showed increased sperm motility and TOSC in men with OAT. In the carnitine-treated patients, nine pregnancies occurred during therapy, and five pregnancies were successful in the combined LC plus LAC treatment group.

In contrast to the aforementioned study, Sigman et al. found no significant positive association between LC and LAC therapy and sperm motility and concentration, with no statistical difference between the two groups [27].

Garolla et al. reported the effect of LC therapy and phospholipid hydroperoxide glutathione peroxidase (PHGPX) therapy in men with OAT [34]. In this study, 30 men with idiopathic OAT were treated in a double-blind study. They formed two groups of patients. One patient group was treated with placebo for 3 months and then 2 gr LC daily for 3 months. LC therapy showed improvements in sperm motility in patients with normal PHGPX levels.

### 3.3. Coenzyme Q10 (CoQ10)

CoQ10 (ubiquinon) is another AS. As an electron transport chain component, it is involved in aerobic cellular respiration, which generates cellular energy compounds such as ATP. This oil-soluble vitamin-like substance has been shown to be present in cell membranes and lipoproteins [62].

Balercia et al. reported the effect of CoQ10 in infertile men on sperm motility. In their study, 60 men showing idiopathic OAT received CoQ10 therapy in a double-blind controlled trial [36]. After 6 months of treatment, CoQ10 content in the ejaculate of patients receiving CoQ10 increased, while sperm motility improved. Six spontaneous pregnancies occurred in the group of patients treated with CoQ10, while three spontaneous pregnancies occurred in the group of patients receiving placebo.

Safarinejad et al. reported 228 infertile men showing abnormal sperm concentration, motility, and morphology in another double-blind controlled intervention [39]. They showed that 26 weeks of treatment with ubiquinone resulted in improvements in sperm density, sperm morphology, and sperm motility in the intervention group compared with the control group in that study.

Nadjarzadeh et al. reported a double-blind, placebo-controlled clinical trial in 47 infertile men with OAT [63]. Patients were randomized to receive 200 mg CoQ10 or placebo daily during 16 weeks of treatment. The study showed no significant changes in semen parameters, such as motility, density, or morphology, in the verum group. On the other hand, total antioxidant capacity was significantly increased after treatment (*p* < 0.05). They showed that three-month AS with CoQ10 increased superoxide dismutase (SOD) and catalase in OAT patients compared to the control group. Furthermore, the authors showed a positive correlation between normal sperm morphology and CoQ10 concentration with additionally increased catalase and SOD concentrations. Additionally, in seminal plasma, 8-isoprostane in both groups (*p* = 0.003), a significant difference was found after supplementation. They concluded that CoQ10 concentration correlates with important semen parameters, such as sperm concentration, motility, and morphology, due to improvements in total antioxidant capacity.

Thakur et al. reported a daily administration of 150 mg CoQ10 improving semen parameters in oligospermic men [64].

In this context, and in contrast to the mentioned studies above, another meta-analysis showed that supplementation of CoQ10 in infertile men does not increase live-birth or pregnancy rates, while CoQ10 showed a general improvement in sperm parameters, such as motility and sperm concentration, as well as CoQ10 concentration in semen [15].

### 3.4. Zinc

Zinc is an abundant metal in the body, second only to iron. The WHO reports that 17.3% of the world’s population is at risk of insufficient zinc intake and that red meat, fish, and milk contain abundant zinc [65]. Studies have shown that zinc supplementation has a protective effect on the spermatozoa against oxidized thiol levels and, therefore, may restore impaired semen function. Alsalman et al. reported on 60 infertile men who received 220 mg zinc sulfate per day for 3 months and found that oxidized thiol levels and semen returned to normal [66]. Zinc is known as a metal that plays an important role in testicular development and sperm maturation [67].

Low sperm zinc concentrations are associated with reduced sperm fertilization capacity [68].

Ebisch et al. reported on men receiving 66 mg of zinc and 5 mg of folic acid for 26 weeks [69]. They reported an improvement in sperm concentration. However, no improvement was observed in other sperm parameters. In contrast to baseline, a positive correlation was found between motility, serum sperm concentration, inhibin B levels, and Zinc.

Hadwan et al. reported the effect of a zinc supplement on sperm characteristics in men with OAT [70]. In this study, 60 fertile and 60 infertile men (age-adjusted) received two zinc sulfate capsules (220 mg per capsule) daily for a period of 3 months. The results showed that total normal sperm count, sperm volume, and percentage of progressive sperm motility increased after zinc therapy. To understand the results of the study in detail, it may be necessary to explain the binding sites of zinc in seminal plasma. In Hadwans’ study, the percentage of high-molecular-weight ligands in sperm was higher in fertile men than in infertile men, showing that zinc supplementation increased the percentage of high-molecular-weight ligands in men with OAT and raised low-molecular-weight ligands to a normal level [71].

Raigani et al. reported a lack of improvement in sperm concentration, motility, and morphology in infertile men with severely impaired sperm parameters after 16 weeks of supplementation with folic acid, zinc, and a combination of these substances [72].

### 3.5. Selenium and N-Acetyl-Cysteine

Selenium has been shown to be an essential trace element for testosterone biosynthesis and formation of sperm [73]. Over 25 selenoproteins have been identified in humans. These selenoproteins contribute to the maintenance of normal sperm structure. N-acetylcysteine is a naturally occurring compound. It is a reaction product of the amino acid L-cysteine and functions as a precursor of glutathione peroxidase. Randomized clinical trials have reported that selenium supplementation alone or with other compounds improves sperm count, motility, and morphology, as well as sperm concentration in infertile men [25,47].

Safarinejad et al. reported the effect of selenium and N-acetyl-cysteine in 468 infertile men with idiopathic OAT [25]. The patients were followed during a 30-week study period. As a result, serum-follicle-stimulating hormone (FSH) decreased while serum testosterone and inhibin B levels increased. As a result, all semen parameters improved significantly in the treated population. In addition, the administration of selenium plus N-acetyl-cysteine further improved semen parameters.

In conclusion, multi-AS showed effective results in male infertility. The synergistic effect of multiple antioxidants makes them interesting for clinicians and andrologists all over the world. Nevertheless, the various studies on multi-AS are inconsistent.

Galatioto et al. reported a study to determine the efficacy of AS therapy on semen parameter quality and the natural pregnancies in infertile men 6 months after retrograde varicocele treatment [33]. Twenty men with varicocele received AS therapy: NAC and vitamin-minerals (vitamin E, vitamin C, vitamin A, thiamin, biotin, B12, riboflavin, magnesium, iron, copper, manganese, zinc). As a result, a significant increase in sperm count was observed in the treated population. In contrast, no significant association was found between multi-AS treatment and other sperm parameters, such as morphology and motility. After 12 months, no spontaneous pregnancy was observed.

Abad et al. reported a study to determine the effect of oral AS treatment on sperm DNA fragmentation (SDF) [22]. In their cohort of 20 infertile patients diagnosed with OAT, all subjects received 1500 mg LC, 60 mg vitamin C, 20 mg CoQ10, 10 mg vitamin E, 10 mg zinc, 200 micrograms folic acid, 50 micrograms selenium, and 1 microgram vitamin B12 for 3 months. The reported results showed that the percentage of DNA-degraded sperm was significantly reduced. Basic sperm outcome showed an increase in the parameter’s concentration, motility, vitality, and morphology. In addition, a significant improvement in DNA integrity was reported. The mentioned paper suggests that AS treatment improves sperm quality, including basic semen parameters and DNA damage, and also helps to ensure DNA integrity and reduce SDF.

Tremellen et al. reported a prospective, double-blind, randomized, placebo-controlled trial of 60 couples with infertile men [48]. Patients were randomly assigned to a treatment group taking a capsule containing 6 mg Lycopene, 500 mg vitamin C, 400 IU vitamin E, 26 micrograms selenium, 25 mg zinc, 5 mg folate, and 1000 mg garlic or placebo daily for 3 months. The antioxidant group had a statistically significant improvement in successful pregnancy rate (38.5%) compared to the control group (16%). In oocyte fertilization rate and embryo quality, no significant changes were observed between both groups.

The main problem with the use of AS is its administration, although there is no strong evidence of benefit from treatment. As shown in large reviews, e.g., by Avila et al. reported in 2022, there are few studies demonstrating consistent benefits of antioxidant therapy, and the evidence is largely controversial. In addition, the available evidence appears to be of low quality, with appropriate analyses in small cohorts [74].

This is also one of the main reasons why European guidelines do not recommend AS for infertile men today. Simenonidis et al. also reported the lack of evidence that changes in semen quality cannot alter the rate of successful pregnancies as the main goal of AS treatment [8].

Sengupta et al. 2022 reported that extensive AS can lead to excess reductants and impede essential oxidation mechanisms in homeostasis, negatively affecting fertility [75].

Sadeghi et al. 2022 also focused on the balance between oxidative and reductive stress with regard to sperm integrity. According to their research, extensive AS can increase reductive stress, which leads to an impaired sperm function and may, therefore, be counterproductive [76].

## 4. Conclusions

In our review, we have shown that the effect of AS therapy in improving male fertility has been extensively studied over the years and worldwide. In some studies, AS was found to be beneficial in reversing OS-related sperm dysfunction and improving pregnancy rates. The most commonly used preparations, either as monotherapy or in combination as multi-AS, were: vitamin E (400 mg), carnitines (500–1000 mg), vitamin C (500–1000 mg), CoQ10 (100–300 mg), NAC (600 mg), zinc (25–400 mg), folic acid (0.5 mg), selenium (200 mg), and lycopene (6–8 mg).

However, the recognition of an ideal AS treatment method is still debatable due to the heterogeneity in study designs and the multifactorial genesis of infertility. Moreover, the studies include different AS regimens at different concentrations. In addition, the normal physiological level of all antioxidants is still unknown, making it difficult to accurately diagnose and choose the best treatment options. Further prospective studies are needed to find the optimal combination of AS used for the treatment of male infertility worldwide. In addition, extrinsic and environmental factors may also be considered in these studies, making a perfect study on this topic complex and difficult.

## Figures and Tables

**Figure 1 antioxidants-12-00836-f001:**
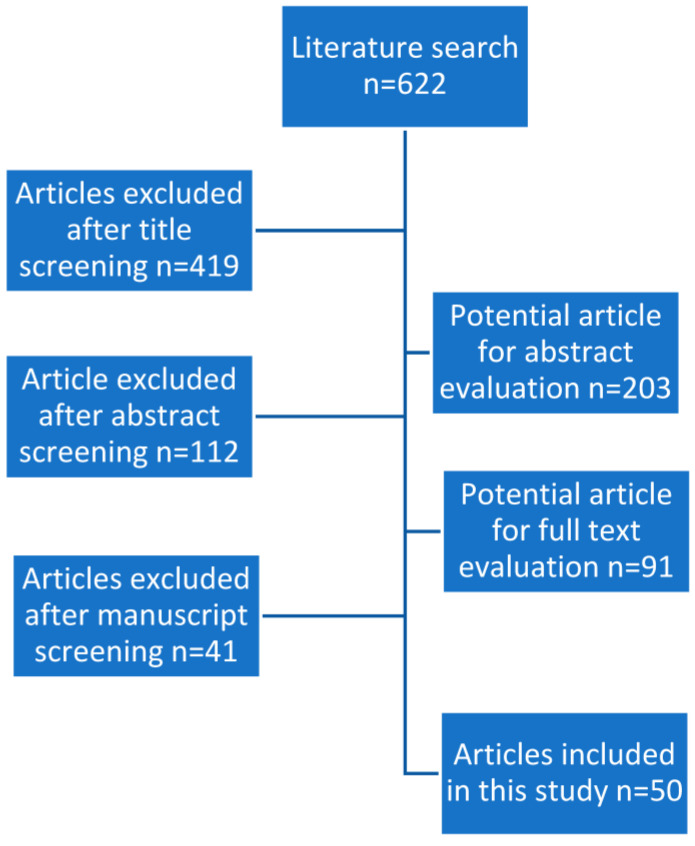
PRISMA flowchart of study selection.

**Table 1 antioxidants-12-00836-t001:** Method of action of widely available antioxidants.

Antioxidant	Mechanism of Action
Ascorbic acid (vitamin C)	Neutralizes free radicals [20]
Enzymatic antioxidants (catalase, peroxidase, etc.)	Neutralizes free radicals [21]
Folate (vitamin B9)	Scavenges free radicals [22]
Tocopherol (vitamin E)	Neutralizes free radicals [23,24]
Selenium	Enzymatic antioxidant activity enhancement [25]
Zinc	Nicotinamide adenine dinucleotide phosphate oxidase inhibition [26]
Carnitines	Neutralizes free radicals; acting as energy source [27]
CoQ10	Reduced form active in humans, scavenges free radicals intermediate in mitochondrial electron transport system [28]
NAC	Supporting enzymatic antioxidant activity [29]
Lycopene	Neutralizing free radicals [30]

**Table 2 antioxidants-12-00836-t002:** List of antioxidants with proposed indication in reviewed evidence (selection).

Diagnosis	Antioxidant	Ref.
Oligozoospemia	2000 mcg of Lycopene twice a day	[30]
	NAC 10 mg/kg/die, Vit C 3 mg/kg/die, Vit E 0.2 mg/kg/die, Vit A 0.06 IU/kg/die, thiamine 0.4 mg/kg/ die, riboxavin 0.1 mg/kg/die, piridoxin 0.2 mg/kg/die, nicotinamide 1 mg/kg/die, pantothenate 0.2 mg/kg/die, biotin 0.04 mg/kg/die, cyanocobalamin 0.1 mg/kg/die, ergocalciferol 8 IU/kg/die, calcium 1 mg/kg/die, magnesium 0.35 mg/kg/die, phosphate 0.45 mg/kg/die, iron 0.2 mg/kg/ die, manganese 0.01 mg/kg/die, copper 0.02 mg/kg/die, zinc 0.01 mg/kg/die	[33]
	LC (2 g)	[34]
	CoQ10 (300 mg)	[35]
	200 μg selenium orally daily, 600 mg N-acetyl-cysteine orally daily, 200 μg selenium plus 600 mg N-acetyl-cysteine orally daily	[25]
	CoQ10 (200 mg)	[36]
	Folic acid (5 mg) + zinc (66 mg)	[37]
	zinc sulphate 200 mg twice daily, zinc sulphate 200 mg + vitamin E 10 mg twice daily, zinc sulphate 200 mg + vitamin E 10 mg + vita- min C 5 mg twice daily	[21]
	NAC (600 mg) and selenium (200 mg)	[35,38,39]
	Lycopene (2 mg)	[30]
	Selenium (200 mug) in combination with vitamin E (400 units)	[40]
	LC 2 g/d and LAC 1 g/d	[27,41,42]
	LC 3 g/d or/and LAC 3 g/d	[42]
	LC (2 g/d) + LAC (1 g/d) + Cinnoxicam (NSAID) 30-mg/d	[43]
Asthenozoospermia	NAC (600 mg) and selenium (200 mg)	[25]
	Lycopene (2 mg)	[30]
	1 g vitamin C and 1 g vitamin E	[44]
	NAC (600 mg)	[29]
	CoQ10 (300 mg)	[35]
	CoQ10 (200 mg)	[36,39]
	l-carnitine (2 g/d) and l-acetyl-carnitine (1 g/d)	[41]
	LC (2 g) and LAC (1 g)	[27,41]
	NAC (600 mg/d orally)	[29]
	LC 3 g/d and LAC 3 g/d	[42]
	Zinc 500 mg/d	[45]
	LC (2 g/d) + LAC (1 g/d) + Cinnoxicam (NSAID) 30-mg/d	[43]
Teratoozoospermia	l-carnitine (2 g/d) and l-acetyl-carnitine (1 g/d)	[41]
	NAC (600 mg) and selenium (200 mg)	[39]
	1 g vitamin C and 1 g vitamin E	[44]
	CoQ10 (200 mg)	[36]
	Lycopene (2 mg)	[30]
	vitamins C and E (400 mg each), β-carotene (18 mg), zinc (500 μmol) and selenium (1 μmol)	[46]
	LC 3 g/d and LAC 3 g/d	[42]
	LC (2 g/d) + LAC (1 g/d) + Cinnoxicam (NSAID) 30-mg/d	[43]
OS	2000 mcg of Lycopene	[30]
	l-carnitine (2 g/d) and l-acetyl-carnitine (1 g/d)	[41]
	Vitamin E (400 mg) and selenium (225 g)	[47]
	Lycopene 6 mg Vitamin E 400 IU Vitamin C 100 mg Zinc 25 mg Selenium 26 µgm Folate 0.5 mg Garlic 1000 mg	[48]
	NAC (600 mg)	[29]
	1 g vitamin C and 1 g vitamin E	[44]
	Vitamin E (600 mg)	[49]
	Vitamin E (400 mg) and selenium (225 g)	[47]
High SDF	Vitamin C (400 mg), vitamin E (400 mg), b-carotene (18 mg), zinc (500 mmol) and selenium (1 mmol)	[46]
	LC (1500 mg); vitamin C (60 mg); CoQ10 (20 mg); vitamin E (10 mg); zinc (10 mg); folic acid (200 lg), selenium (50 lg); vitamin B12 (1 lg)	[22,50]
	Vitamin E 100 mg	[51]
Improving perm function tests	Vitamin E (600 mg/d)	[49]
Improving success rate of ART	Vitamin C (1 g) + vitamin E (1 g)	[52]
	Vitamin E 600, 800 or 1200 mg (no improvement in sperm parameters)	[53]
	Selen (300 microg/d) no improvement on standard semen parameters.	[54]
	Lycopene (6 mg), vitamin E (400 IU), vitamin C (100 mg), zinc (25 mg), selenium (26 g), folate (0.5 mg) and garlic (1 g)	[48]
	Zinc 500 mg improvement in standard sperm parameters	[45]
	CoQ10 300 mg orally twice daily for 12 months	[35]
Live-birth rate	Vitamin E 600 mg/d improved zona binding assay	[49]
	LC (2 g) + LAC (1 g/day) + 30-mg cinnoxicam (NSAID)	[43]
	Vitamin E 100 mg	[51]
	group 3: L-carnitine/acetyl-L-carnitine + 1 × 30-mg cinnoxicam (NSAID), L-carnitine (2 g/d) + acetyl-L-carnitine (1 g/d)	[43]
	Vitamin E (400 mg) and selenium (225 g)	[47]
	Vitamin E 100 mg	[51]
	Vitamin C (1 g) + vitamin E (1 g)	[52]

**Table 3 antioxidants-12-00836-t003:** Preparations with potential positive effect on male infertility.

Substance	Dosage
vitamin E	400 mg
carnitines	500–1000 mg
vitamin C	500–1000 mg
CoQ10	100–300 mg
NAC	600 mg
zinc	25–400 mg
folic acid	0.5 mg
selenium	200 mg
lycopene	6–8 mg

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
