# Peer review of "Antioxidant Supplementation on Male Fertility—A Systematic Review"

_antioxidants, 2023, doi:10.3390/antiox12040836_

Round 1

Reviewer 1 Report

This Review study is based on the effects of antioxidant supplementation to semen on some selected sperm parameters. What is the rational of this Review? This Review topic has been extensively covered in a lot of papers. How could you write a Review paper without contributing to the studied field? I  can not find any published paper of the Authors in the References.

The Reviewer suggests that the following comments would be helpful to improve the quality of the manuscript.

1. Rewrite the objective of the study (L71-72). It is a bit too confusing.

2. Unclear about the significance of Table.1. Missing are the enzymatic antioxidants.

3. The data shown in Table 2 are unreadable and do not correspond to the objective of the Review. Select the most common antioxidants. Also, it is difficult to follow the flow chart shown in Figure 1.

Author Response

This Review study is based on the effects of antioxidant supplementation to semen on some selected sperm parameters. What is the rational of this Review?

We aim to present an up to date review including the latest evidence published.

This Review topic has been extensively covered in a lot of papers. How could you write a Review paper without contributing to the studied field? I  can not find any published paper of the Authors in the References.

Prof. Dimitriadis is a known expert in the filed of andrology and published extensively in this field.

The Reviewer suggests that the following comments would be helpful to improve the quality of the manuscript.

  1. Rewrite the objective of the study (L71-72). It is a bit too confusing.

We changed the manuscript accordingly.

  1. Unclear about the significance of Table.1. Missing are the enzymatic antioxidants.

We added enzymatic antioxidants to the table

  1. The data shown in Table 2 are unreadable and do not correspond to the objective of the Review. Select the most common antioxidants. Also, it is difficult to follow the flow chart shown in Figure 1.

As the AS given in the mentioned studies are very complex it requires to include them into the table. Perhaps we should put table 2 into the supplement data, so the main text will be easier to read for a quick review. If the reader likes to go more into detail, table 2 is still available for studying the details.

Reviewer 2 Report

In the review titled “Antioxidant Supplementation on Male Fertility-A systematic review” the authors review the current literature regarding the effect of antioxidant supplementation on male fertility parameters.

 This review lacks some useful information from the literature. Therefore, I think that I can reconsider the possibility of publication after a major revision.

In table 1 it is necessary to add the reference of the papers that demonstrate the way of action of the various substances

 Page 11 has instructions but missing author's statements for various items

In my opinion, in the introduction, the following recent works must be added, regarding the Mediterranean Diet as a Shield against Male Infertility and Cancer Risk Induced by Environmental Pollutants:

10.3390/ijms23031568

 damage induced by heavy metals, for example, can be protected by the antioxidant activity of different medicinal plants such as Fejioa selloniana for which antimicrobial and antioxidant activity has been described in literature.

The authors must better argue on the decline of male fertility worldwide. In the last decade environmental pollution has a strong impact on semen quality.

By now, the spermiogram, is not sufficient to define the fertilizing capacity of semen. Molecular investigations are needed. Argue on this point considering the effects of pollution on semen, quoting some recent works which indicated that there are molecular alterations and severe abnormalities in spermatozoa of young men living in polluted areas as reported in this work: 10.3390/ijerph191711023

 In addition, I should  emphasized that in Highly Polluted Areas in Italy Macro and Trace Element Concentrations were found in Human Semen and Blood Serum.

Moreover, a decrease antioxidant activity in semen has been found in people living in area at high environmental impact and possible transgenerational effects of pollutants have been described in polluted areas regarding the molecular alterations in spermatozoa.

Another point that must be better discussed is relative to DNA oxidative damage.

Oxidative damage is a major cause of infertility. There is a lot of work showing the correlation between certain heavy metals and oxidative damage to DNA. In addition, it has been demonstrated that that some heavy metals can change the properties of sperm nuclear basic proteins. In fact, in areas of high environmental impact sperm nuclear basic proteins can change their canonical protective role towards DNA and be involved in oxidative DNA damage

 Semen has been for some time considered an early sentinel of the health status of the environment and also of the general health of an individual. It is very strange that the authors did not find in their research some works related to the effects of pollutants on human reproductive health that are part of the Ecofoodfertility project that in this review is not mentioned at all.

 I believe that to make the review more appealing you should include a nice color image that summarizes the results and the message the authors want to give

Author Response

In the review titled “Antioxidant Supplementation on Male Fertility-A systematic review” the authors review the current literature regarding the effect of antioxidant supplementation on male fertility parameters.

This review lacks some useful information from the literature. Therefore, I think that I can reconsider the possibility of publication after a major revision.

Thank you for your consideration.

In table 1 it is necessary to add the reference of the papers that demonstrate the way of action of the various substances.

We added some references to table 1.

 Page 11 has instructions but missing author's statements for various items

  We revised this section.

In my opinion, in the introduction, the following recent works must be added, regarding the Mediterranean Diet as a Shield against Male Infertility and Cancer Risk Induced by Environmental Pollutants:

10.3390/ijms23031568

 damage induced by heavy metals, for example, can be protected by the antioxidant activity of different medicinal plants such as Fejioa selloniana for which antimicrobial and antioxidant activity has been described in literature.

We added the mentioned study.

The authors must better argue on the decline of male fertility worldwide. In the last decade environmental pollution has a strong impact on semen quality.

We added one passage into the introduction on this matter.

By now, the spermiogram, is not sufficient to define the fertilizing capacity of semen. Molecular investigations are needed. Argue on this point considering the effects of pollution on semen, quoting some recent works which indicated that there are molecular alterations and severe abnormalities in spermatozoa of young men living in polluted areas as reported in this work: 10.3390/ijerph191711023

 In addition, I should  emphasized that in Highly Polluted Areas in Italy Macro and Trace Element Concentrations were found in Human Semen and Blood Serum.

We added that publication and emphasized this in one passage in the introduction.

Moreover, a decrease antioxidant activity in semen has been found in people living in area at high environmental impact and possible transgenerational effects of pollutants have been described in polluted areas regarding the molecular alterations in spermatozoa.

Another point that must be better discussed is relative to DNA oxidative damage.

Oxidative damage is a major cause of infertility. There is a lot of work showing the correlation between certain heavy metals and oxidative damage to DNA. In addition, it has been demonstrated that that some heavy metals can change the properties of sperm nuclear basic proteins. In fact, in areas of high environmental impact sperm nuclear basic proteins can change their canonical protective role towards DNA and be involved in oxidative DNA damage

 Semen has been for some time considered an early sentinel of the health status of the environment and also of the general health of an individual. It is very strange that the authors did not find in their research some works related to the effects of pollutants on human reproductive health that are part of the Ecofoodfertility project that in this review is not mentioned at all.

We thank the reviewer about this comment. Nevertheless, the aspect of environmental impact on fertility is not the main aspect of our review. We wanted to focus on AS and adding more literature on the important aspect of pollution may change the focus of the review. It it also to important to just be a small passage mentioned beside. We would emphasis to write a separate review on this very important aspect about environmental pollution and fertility.

 I believe that to make the review more appealing you should include a nice color image that summarizes the results and the message the authors want to give.

We added a new table 3 for better visualization of beneficial AS

Round 2

Reviewer 1 Report

The Authors have addressed my comments except comment "This Review topic has been extensively covered in a lot of papers. How could you write a Review paper without contributing to the studied field? I cannot find any published paper of the Authors in the References".

Prof. Dimitriadis is a known expert in the filed of andrology and published extensively in this field.

Please include papers of prof. Dimitriadis in the References of the review paper to confirm your statement.

Author Response

https://pubmed.ncbi.nlm.nih.gov/36194789/

https://pubmed.ncbi.nlm.nih.gov/34356300/

https://pubmed.ncbi.nlm.nih.gov/33563149/

We will add this to our review.

Reviewer 2 Report

Accept in present form

Author Response

Thank you.